# HuD regulates apoptosis in N2a cells by regulating Msi2 expression

**Naina Gaikwad**[1,2]**, Rucha Sarwade**[1,2]**, Sourav Halder**[1,2]**, Gaurav Agarwal**[1,2]**, Vasudevan Seshadri**[1] *

1 National Centre for Cell Science, Ganeshkhind, Pune, India, 2 Savitribai Phule Pune University, Ganeshkhind, Pune, India

* seshadriv@nccs.res.in

## Abstract

HuD plays a critical role in neurite outgrowth, neuronal plasticity, and survival. However, HuD autoantibodies from patients with paraneoplastic gut dysmotility can trigger the apoptotic cascade in human neuroblastoma cell line and myenteric neurons. The mechanism by which HuD regulates the apoptotic pathway is unclear. Apoptosis is one of the underlying causes of neurodegenerative diseases like Alzheimer's disease. In the current study, we found that HuD interacts with *Msi2* transcript and positively regulates it in the mouse neuroblastoma (N2a) cells. MSI2 being an RNA binding protein has diverse mRNA targets and regulates the mitochondrial apoptotic pathway by interacting with and repressing APAF1 transcript. Conversely, the reduced levels of HuD leads to decreased Msi2 expression and increased APAF1 levels, which results in apoptosis in N2a cells. Overall, our research indicates that HuD and Msi2 possess an anti-apoptotic role in N2A cells.

## Introduction

HuD is a member of Hu/ELAV family of RNA binding proteins, known for its predominant expression in neurons. It is specifically expressed in neuronal subtypes such as motor, sensory, and hippocampal neurons of humans and mice [1, 2]. HuD is also known to be expressed in both the pancreatic α cells of mice and β cells of both mice and humans [3, 4]. This multifunctional RNA binding protein plays an important role in various cellular processes, including alternative splicing, alternative polyadenylation, translation regulation, RNA translocation, and RNA stability [5–8]. HuD can regulate mRNA expression by identifying and binding to AU-rich elements on target mRNAs [9].

Neuronal HuD regulates multiple targets to carry out its functions such as neural differentiation, neurite extension, and neuronal plasticity [2, 10–14]. Additionally, it also regulates other RNA binding proteins like *Msi1* and *NOVA1*, thereby amplifying the impact of its gene regulatory network [15, 16]. Human neuroblastoma cells exposed to the sera of paraneoplastic gut dysmotility disease patients, containing HuD autoantibodies, undergo mitochondrial apoptosis [17]. Further, HuD knockdown under basal conditions in βTC6 cells leads to an increase in the cleavage of caspase 3, leading to apoptosis. However, exposure of cells to cytokines under

**Data Availability Statement:** All relevant data are within the manuscript and its Supporting Information files.

**Funding:** The author(s) received no specific funding for this work.

**Competing interests:** The authors have declared that no competing interests exist.

HuD knockdown conditions prevents apoptosis [18]. Decreased HuD levels is found to affect its interaction with ADAM10 mRNA, an α-secretase that is responsible for the formation of soluble amyloid β peptides (Aβ) in human neuroblastoma cells [19]. This study also shows that an increase in $Aβ_{1-42}$ causes a reduction in nELAV levels in the AD hippocampi. HuD is known to stabilize the mRNA of Neprilysin, which is responsible for the degradation of Aβ peptides [8]. However, in contrast to these findings, HuD is also known to promote Aβ accumulation by stabilizing the APP, BACE, and Neuroserpin RNAs in AD patient brains [20–23]. These studies indicate a significant role of HuD in neural differentiation, apoptosis, and Alzheimer's disease.

Msi2 is a member of the Musashi family of RNA binding proteins and is characterized by two RNA recognition motifs [24, 25]. It is expressed in neural progenitor cells and specific neuron populations such as GABAergic and hippocampal neurons [24]. Msi2 preferentially binds to UUAG, UAG, and U-rich sequences and targets developmental transcription factors, cell cycle regulators, and cell survival genes [26]. Knockout of Msi2 in human leukemic cells (K562) induced apoptosis and thus prevents leukemic cell proliferation [27]. Msi2 is upregulated in various cancers such as myeloid leukemia, cholangiocarcinoma, and pancreatic cancer [28–30]. These studies indicate a role for Msi2 in the maintenance of progenitor cells, cell proliferation and apoptosis.

Apoptosis is the process of programmed cell death that occurs during early stages [31]. Several factors regulate cellular apoptosis and any dysregulation of these factors can initiate apoptotic cascade leading to untimely cell death [32]. Apoptosis can be a major contributor to cell death in several disease conditions like that of Alzheimer's disease [33–36]. These findings highlight the critical importance of tightly regulating the molecules involved in apoptosis.

Several studies suggested a role for HuD and Msi2 in various cellular processes including apoptosis, however some of the molecular details are yet to be elucidated. Pulldown, followed by microarray were utilized to determine the in vivo mRNA targets of HuD in the mouse forebrain [37]. Msi2 mRNA was one of the targets transcripts of HuD. To validate the interaction and to understand the functional relevance of this interactions between HuD and Msi2 mRNA, we carried out RNA immunoprecipitation and functional assays in N2a cells. We observed that HuD could interact with *Msi2* RNA as well as regulate its RNA and protein levels. Using the online database POSTAR, we identified the target of Msi2, i.e. APAF1, which is involved in the mitochondrial apoptotic pathway. APAF1 is the rate limiting factor in apoptosis, as high levels of APAF1 can promote apoptosis and reduced expression of APAF1 delays mitochondrial damage and significantly reduce apoptosis [38–41]. Our findings suggest that Msi2 binds to APAF1 mRNA and down-regulates it, ultimately influencing the process of mitochondrial apoptosis.

## Materials and methods

### Cell culture

N2a cells were obtained from the NCCS cell repository, the cells were free of mycoplasma and validated. Early passage cells were maintained in DMEM 2 g Glucose/L (Himedia), 1.5 g/L Sodium Bicarbonate, 4mM Glutamax media supplemented with 10% Fetal Bovine Serum (MP Biomedicals), 100 units/ml penicillin and 100μg/ml streptomycin (Invitrogen). Cells were maintained at 37˚C at 5% $CO_2$.

### U. V crosslinking of adherent cells

After cells reached the required confluency, the media was taken out and twice-washed with PBS. The culture plates were stored on ice without a lid and subjected to three cycles of UV

exposure at an energy of 120mJ/cm$^2$ for 2 minutes each cycle. Following cross-linking, cells were PBS-washed and scraped the pellet was quickly frozen in liquid nitrogen and kept at -80°C.

## RNA immunoprecipitation

RIP assay was performed as explained in the previous study [7]. Cell supernatant was pre-cleared with protein G beads (G biosciences) for 1 hour. This lysate (750μg) was subjected to immunoprecipitation with respective antibodies 4μg of rabbit anti-HuD (sc-25362) and Msi2 (ab76148) at 4°C for 16 hours in the presence of 1X protease inhibitor, 1mM PMSF and 20U of RNAsin. Rabbit IgG was used as a negative control. An appropriate amount of the protein G beads (G Biosciences) was added to the protein antibody conjugates and was incubated for 4 hours at 4°C. RT-PCR analysis of Msi2 in HuD pulldown and that of APAF1 in Msi2 pulldown was performed on extracted RNA. 1/10$^{th}$ volume of IP supernatant was also used for protein precipitation assessment by western blotting using respective antibodies.

## Cloning of Msi2 in peGFPC1 mammalian expression vector

N2a cells showed maximum expression of Msi2, hence RNA from this cell line was used for amplification of Msi2. RNA was extracted from a single 100 mm dish containing N2a cells using trizol. The RNA was quantified using nanodrop and 1μg of RNA was used for cDNA synthesis using oligo dT primer. cDNA from N2a cells were used as a template for PCR using Msi2 cloning forward primer (GCGCAGATCTATGGAGGCAAATGGGAGCC), Msi2 cloning reverse Primer (GCGCGTCGACTCAGTGTATCCATTTGTAAAGGCC) containing the restriction sites for the enzymes BglII and SalI respectively. The PCR product was purified and then the PCR products as well as the vector (pEGFP-C1) were subjected to digestion using the enzymes BglII and SalI. The digested samples were gel purified and a ligation reaction was set up using T4 ligase enzyme (Thermo Fisher Scientific) at 16°C overnight. The ligated products were then transformed into *E. Coli* DH5α. The transformants were screened by colony lysis, colony PCR and restriction digestion. The clones were also confirmed using sequencing.

## Transient transfection of plasmids and siRNAs

N2a cells were seeded to obtain a cell density of 70% on the day of transfection. Before transfection, the complete media of the cells was replaced by media containing 1%FBS. The previously cloned construct of pcDNA3.1 HA HuDA (full-length protein) was used for overexpression studies [7]. For a 100mm dish, 8 μg of pcDNA3.1 HA-EV and HA-HuDA /peGFPC1 EV and Msi2 constructs were added to 400μl of 150mM NaCl and 80μl of PEI (1:8 ratio). The tubes were slightly tapped for mixing and spun, and the mixture was kept at room temperature for 20–30 minutes. The mixture was further added to the culture for transfection. After 16 hours of transfection, the media of the cells was changed. After 48 hours of transfection, the cells were collected for further assays. To carry out the knockdown of HuD and Msi2 in N2a cells the commercially available siRNAs of HuD (sc-37836) and Msi2 (SASI_MM01_0019-) were used. Once the cells reached 70% confluency the complete media was replaced by 1% FBS-containing media. For a 60mm dish, 60nM of HuD/Msi2 siRNAs were added to 150μl OptiMEM in one tube, and 20μl of Lipofectamine 2000 was added to 150μl of OptiMEM in another. The tubes were gently tapped and spun and kept at room temperature for 5 minutes. After that the contents of both the tubes were gently mixed and the transfection mix was kept at room temperature for 20–30 minutes. The transfection mix was then added to the media in a dropwise manner. The media of the cells was changed after 16 hours and after 48 hours of transfection, the cells were collected.

## Real-Time PCR

To assess the effect of overexpression and knockdown of HuD and Msi2 on its target mRNAs we carried out Real-Time PCR. Primers specific for genes i.e. Msi2 3' UTR FP (GCAGGCG CTTCCATTGCCG), Msi2 3' UTR RP (CAGCAGGTAGCCGATGGGTG), APAF1 3'UTR FP (TGTTTACAGCTGGGCCAGTGG), APAF1 3' UTR RP (GCCAACCAATTTGAATGAATCAA) were used, 18s FP (CGCCGCTAGAGGTGAAATTC), 18s RP (CATTCTTGGCAAATGCTTTCG), was used as a housekeeping gene, the abundance of each gene was analysed using 2x SyBr green mix (Biorad). The final mRNA levels of the gene of interest were determined by normalizing the Ct values of the genes to that of the housekeeping gene 18s rRNA.

## Flow cytometry

To study the phenomenon of apoptosis in N2a cells, flow cytometry was carried out. After the knockdown of HuD/ Msi2 in the N2a cells the cells were washed with 1X PBS and then the cells were trypsinised. Complete media was added to the trypsinised cells and the cells were centrifuged at 1500 rpm for 10 minutes. The supernatant was decanted and the cells were washed with 1X PBS and spun at 1500 rpm for 10 minutes, this step was carried out twice. To the pellets, 100 μl of 1X binding buffer was added and the tubes were gently tapped to mix. To it, 5μl of Annexin V conjugated FITC (CST # 4894S) was added along with 100mg/ml of 5μl of Propidium Iodide. The tubes were gently tapped for mixing. Then the tubes were kept at room temperature in the dark for about 15–20 minutes and then the stained cells were transferred to the FACS tubes for acquisition.

## Western blotting

Whole-cell lysates were prepared using RIPA lysis buffer containing a 1× protease inhibitor, separated on the appropriate per cent of SDS polyacrylamide gel for about 1.5 to 2 hours at 100 volts. The proteins were then transferred onto polyvinylidene difluoride membranes (Millipore). The membranes were blocked with appropriate blocking agents depending on the antibody either 1% Milk/ 1% BSA/ 5% BSA for 1 hour. Then the PVDF membrane was incubated with primary antibodies against HuD (sc-48421, dilution 1:1000), Msi2 (PA5-31024, dilution 1:1000), APAF1 (CST 4452, dilution 1:1000), Bax (CST 2772, dilution 1:1000), Bcl2 (SC-7382, dilution 1:1000) and Tubulin (T-5161, dilution 1:10000) or GAPDH (6004–1, dilution 1:10000) and kept at 4°C rocker overnight. After this, the PVDF membrane was washed thrice with TBST and further incubated with the appropriate secondary antibodies conjugated with horseradish peroxidase (dilution 1:1000). Protein bands were detected using enhanced chemiluminescence (Cynagen) on the Amersham 600 machine.

## Web tools

We used POSTAR2 experimental database for our study. We searched for the mouse species in the CLIP database showing the mRNA that can interact with Msi2. The database identified 17 binding sites for Msi2 on *APAF1* mRNA.

## Statistical analysis

Every experiment has been repeated a minimum of three times. The results were expressed as the average of at least three independent experiments. The standard error of the mean is indicated by the error bars. Graph Pad Prism software version 7 was used to do a two-tailed unpaired t-test on the data. The asterisk (*) denote the p value's (**** = p$\leq$ 0.0001, *** = p$\leq$ 0.001, ** = p $\leq$0.01, * = p $\leq$ 0.05and ns (non-significant) = p > 0.1).

### Ethical approval

We did not use any animals for the experiment, hence, not applicable

## Results

### HuD interacts with *Msi2* mRNA and upregulates its expression

HuD is known to interact with and modulate the expression of certain RNA-binding proteins [15, 16]. To investigate the interaction between HuD and the RNA binding protein Msi2 in N2a cells, we performed an RNA immunoprecipitation assay where HuD protein was pulled down from N2a cell lysates, with IgG serving as a negative control. RT-PCR analysis of the RNA that was captured in the pulldown revealed a significant enrichment of Msi2 in the HuD sample, indicating a specific interaction between HuD and *Msi2* mRNA in these cells (Fig 1A).

To further elucidate the significance of this interaction, we ectopically expressed HuD in N2a cells and examined the transcript levels of *Msi2*. Our results indicated a significant increase in *Msi2* RNA levels in the HuD overexpression group as compared to the vector control (Fig 1B). In agreement with these findings, we found decreased protein and RNA levels of Msi2 upon HuD knockdown (Fig 1C, and S1 Fig). Taken together, these results suggests that HuD interacts with Msi2 mRNA and positively regulates its expression in N2a cells.

### Msi2 interacts with APAF1 mRNA and regulates its expression

Msi2, a versatile RNA binding protein, has been shown to target several mRNA molecules that play important roles in various cellular processes [24]. To identify the mRNA targets of Msi2, we performed in-silico analysis using POSTAR2 web tools which identified APAF1 as potential mRNA target of Msi2,. To confirm the interaction between Msi2 and APAF1 mRNA, we performed Msi2-RNA immunoprecipitation (RIP) using N2a cell lysates. Our results showed a specific enrichment of band corresponding to APAF1 mRNA in the Msi2-RIP lane (Fig 2A), while the IgG control pull-down did not show significant amplification, indicating that Msi2 protein interacts with APAF1 mRNA.

To investigate the functional consequences of this interaction, we overexpressed Msi2 in N2a cells and assessed the RNA levels of *APAF1*. We found a significant decrease (Fig 2B) in *APAF1* transcript levels which was also correlated with reduced protein levels upon Msi2 over-expression in N2a cells (S2 Fig). Conversely, we found elevated APAF1 protein expression upon Msi2 knockdown (Fig 2C and 2D). These experimental results suggest that Msi2 negatively regulates APAF1 expression by interacting with its mRNA.

### *Msi2* knockdown causes an increase in apoptosis in N2a cells

Msi2 is an RNA-binding protein that plays a critical role in regulating cell survival and proliferation [25, 27]. To further understand the physiological effects of Msi2 in neuronal system, we conducted Msi2 knockdown in N2a cells. Using FACS analysis, we observed a significant increase in the percentage of apoptotic cells upon Msi2 knockdown (Fig 3A–3C). These results suggest that Msi2, has an anti-apoptotic role in N2a cells, and its decreased expression leads to increased apoptosis. To further confirm this, we analyzed the Bax/Bcl2 ratio, which is an indicator of the activation mitochondrial apoptotic pathway [42]. We found a significant increase in the Bax/Bcl2 ratio (Fig 3D) upon Msi2 knockdown in N2a cells, indicating an increase in apoptosis in these cells.

**Fig 1. HuD binds to Msi2 mRNA and positively regulates Msi2 transcript in N2a cells.** N2a cells were exposed to UV to crosslink interacting proteins and RNAs and these lysates were used to study the interaction between HuD and Msi2 transcript. **(A)** Western blot of immunoprecipitated samples using N2a lysates probed with HuD antibody (left panel), RT-PCR analysis of HuD immunoprecipitated samples using primers specific for Msi2. The samples used as template for the PCR is indicated, with NRT (No reverse transcriptase) serving as a negative control (Right panel). **(B)** Bar graph represents the relative levels of Msi2 RNA after normalizing with 18srRNA in control (pcDNA3 HA- EV) and HuD over expressing cells (pcDNA3 HA- HuD). RNA levels were analyzed by Real-Time quantitative PCR using gene specific primers. The data represents an average of three independent experiments **(C)** Western blot shows a decrease in the protein levels of Msi2 upon HuD knockdown and bar graph represents the quantitation of the Msi2 protein normalized to GAPDH levels from three independent Western blot (lower panel).

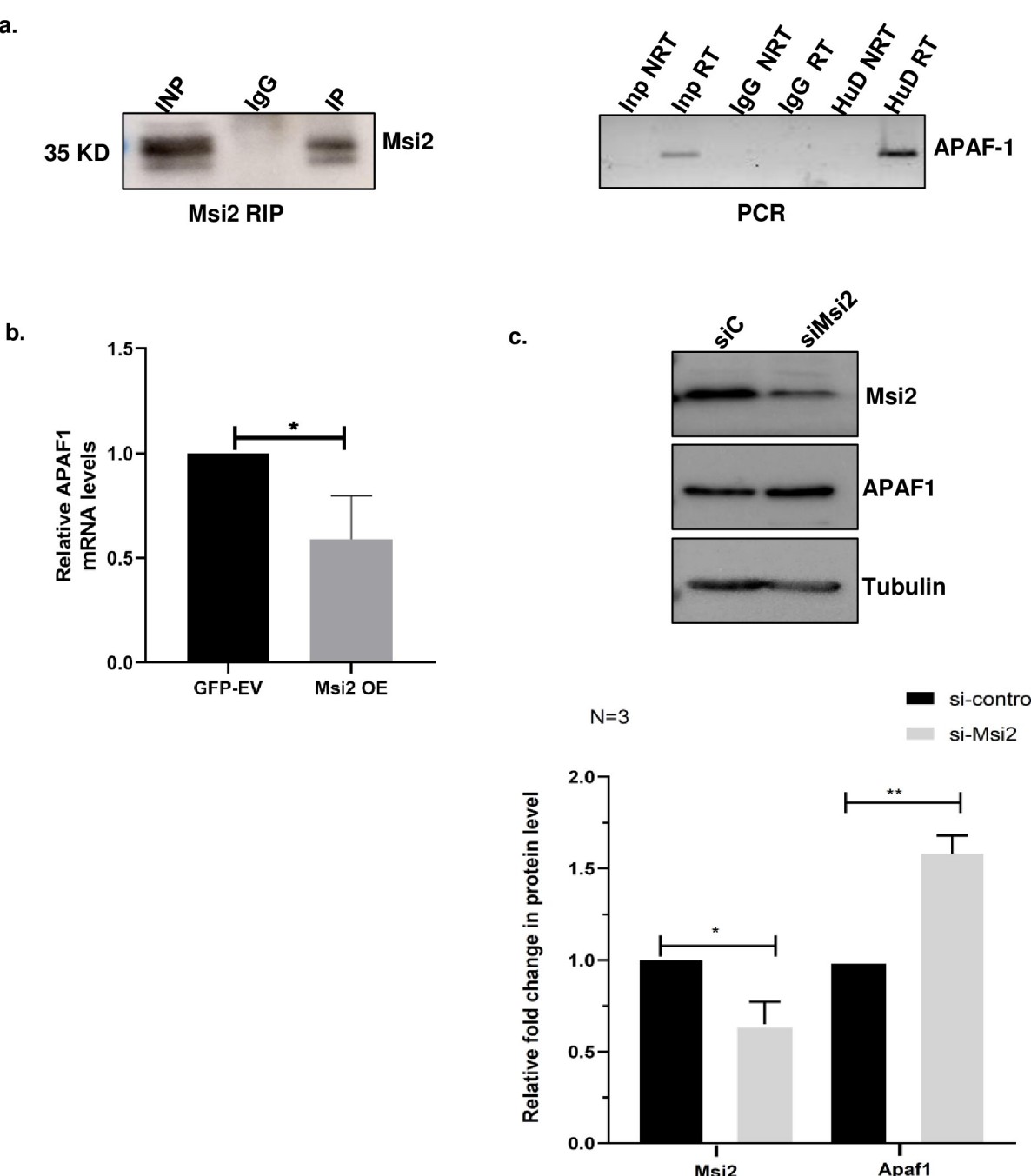

**Fig 2. Msi2 binds to the RNA of APAF1 and negatively regulates its transcript in N2a cells.** Msi2 was pulled down from the N2a cell lysates, and the RNAs associated with Msi2 were analyzed by RT-PCR using specific primers for APAF1. (**A**) Western blot probed with Anti Msi2 antibody to assess Ms2 immunoprecipitation using N2a lysates (Left panel), RT-PCR analysis of immunoprecipitated samples indicates Msi2 association with APAF1 RNA in N2a cells (Right Panel). (**B**) pEGFP.C1 EV and pEGFP.C1 Msi2 constructs were transfected in N2a cells and the RNA levels of APAF1 were analyzed by Real-Time PCR. Relative change in the APAF RNA levels normalized to 18s rRNA are represented in bar graph from three independent experiments and it was found to be increased. (**C**) Western blot analysis of APAF1 in cells transfected with control or Msi2 specific siRNA shows an increase in the protein levels of APAF1 upon Msi2 knockdown and the lower panel shows the bar graph representing the quantitation of the APAF protein normalized to Tubulin levels from three independent western blot.

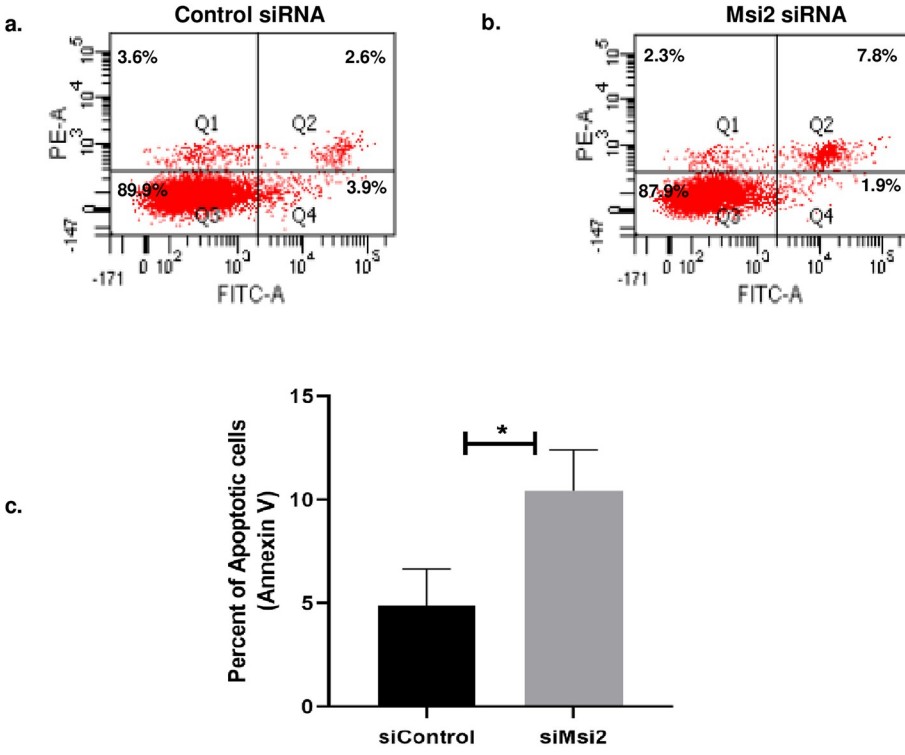

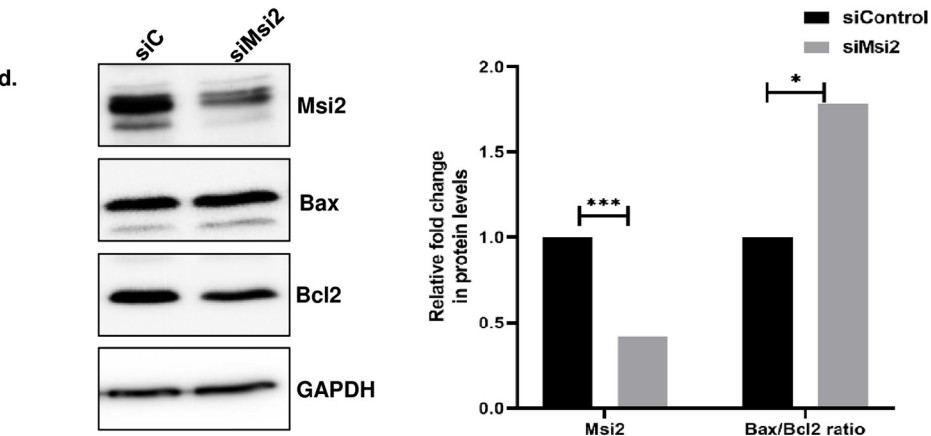

**Fig 3. Knockdown of Msi2 induces apoptosis in N2a cells.** Control siRNA and Msi2 siRNA were transfected in N2a cells, and after 48 hours of transfection cells were trypsinised and stained with Annexin-V conjugated FITC and Propidium Iodide. Cells were then analyzed using flow cytometry (**A**) control siRNA-treated cells and (**B**) Msi2 siRNA-treated N2a cells showed an increase in apoptosis compared to control siRNA (**C**) Bar graph represents the percentage of annexin positive (apoptotic) cells in indicated samples. (**D**) Western blot probed with Msi2, Bax, Bcl2 and GAPDH showing an increase in the ratio of Bax/Bcl2 upon Msi2 knockdown (left panel) and the right panel shows the bar graph representing the relative change in Bax/Bcl2 protein ratio and Msi2 levels normalized to GAPDH levels between control and Msi2 knockdown cells from three independent western blots.

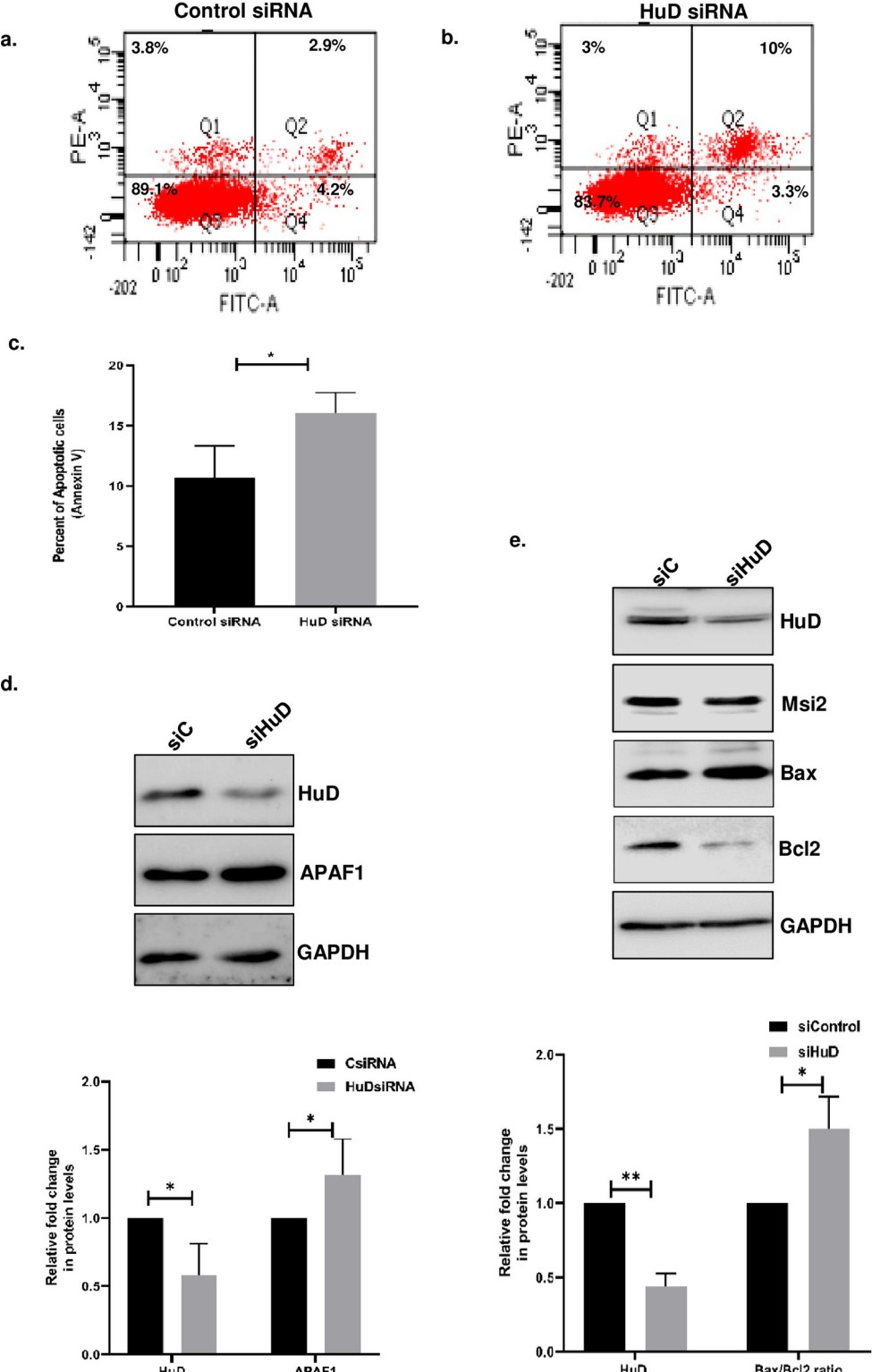

**Fig 4. Knockdown of HuD induces apoptosis in N2a cells.** Control siRNA and HuD siRNA were transfected in N2a cells after 48 hours of transfection cells were trypsinised and stained with Annexin-V conjugated FITC and Propidium Iodide. Cells were then analyzed using flow cytometry (Canto) **(A)** control siRNA-treated cells and **(B)** HuD siRNA-treated N2a cells showed increased apoptosis compared to control siRNA cells. **(C)** Bar graph represents the percentage of annexin positive (apoptotic) cells in indicated samples from three independent experiments. **(D)**

Western blot showing the increase in the levels of APAF1 protein upon HuD knockdown. The lower panel shows the bar graph representing the relative change of APAF and HuD protein normalized to GAPDH levels between the control and HuD siRNA transfected cells from three independent experiments. (**E**) Western blot probed with Msi2, Bcl2, Bax, HuD and GAPDH showing a decrease in the ratio of Bax/Bcl2 upon HuD knockdown. The lower panel shows the bar graph representing the relative change in Bax/Bcl2 protein ratio and HuD levels normalized to GAPDH levels between control and HuD knockdown cells from three independent Western blots.

## *HuD* knockdown causes an increase in apoptosis in N2a cells

HuD is an essential RNA-binding protein that plays a critical role in neuronal survival and differentiation [2, 14]. We investigated the impact of HuD knockdown on N2a cellular apoptosis using FACS. Our results showed that HuD knockdown increased apoptosis in N2a cells (Fig 4A–4C). In accordance with our FACS data, we found significantly increased APAF1 protein upon HuD knockdown in N2a cells (Fig 4D). To further investigate the apoptotic pathway activated upon HuD knockdown we measured the Bax/Bcl2 ratio and found a significant increase in the Bax/Bcl2 ratio (Fig 4E), indicating the activation of the mitochondrial apoptotic pathway upon HuD knockdown in N2a cells.

## HuD and Msi2 act coordinately to prevent cellular apoptosis

Having established the role of HuD and Msi2 in controlling the N2a cell death, we conducted a rescue experiment to determine whether HuD and Msi2 regulate the apoptotic pathway in a synchronized manner. To this end, we performed knockdown of HuD and overexpression of Msi2 simultaneously in N2a cells, followed by FACS analysis to assess changes in cell viability. We found a significant decrease in apoptosis compared to the control siRNA and Empty vector-transfected cells, indicating that both proteins work together to regulate cell survival (Fig 5A–5C). Additionally, we examined the levels of APAF1 protein in N2a cells and found a decrease in its level, while no significant change in the Bax/Bcl2 ratio (Fig 5D) was observed, indicating the concordant gene regulatory network between HuD and Msi2. We also performed a double knockdown of HuD and Msi2 in N2a cells and analyzed the levels of APAF1 protein and Bax/Bcl2 ratio. We observed that the increase in APAF1 protein levels and Bax/Bcl2 ratio was comparable to that of a single knockdown of HuD or Msi2 and no additive increase was observed, (Supp Fig 3). In addition, when we sequentially overexpressed HuD followed by Msi2 again we observed a significant decrease in Bax/Bcl2 ratio, indicating a reduction in cellular apoptosis (S4 Fig). These results clearly suggest a common mode of regulation and that both the proteins work together to regulate the mitochondrial pathway of apoptosis.

## Discussion

HuD is a versatile RNA binding protein that regulates several neuronal processes such as neuronal differentiation, plasticity, and survival [2, 14]. In Paraneoplastic gut dysmotility, autoantibodies targeting the HuD protein trigger the mitochondrial apoptotic pathway in myenteric neurons [17]. On the other hand, Msi2 is a crucial RNA-binding protein that primarily contributes to maintaining specific neuronal lineages [24]. Knocking down Msi2 induces apoptosis in leukemic cells [27]. As HuD and Msi2, both RNA binding proteins play a significant roles in cell death, a tight regulation of these is critical to cell survival an proliferation. The mitochondrial apoptotic pathway is one of the major pathways of programmed cell death, which involves the permeabilization of the outer mitochondrial membrane and the release of cytochrome c to form apoptosome which requires APAF1 [38]. It has been observed that an increase in APAF1 levels can lead to an increase in apoptosis [41–45]. APAF1 expression is generally low in cells, making it a limiting factor in apoptosome formation [39], and regulation

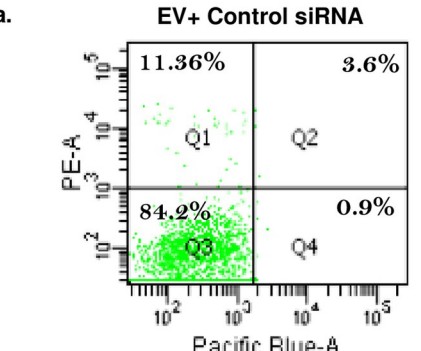
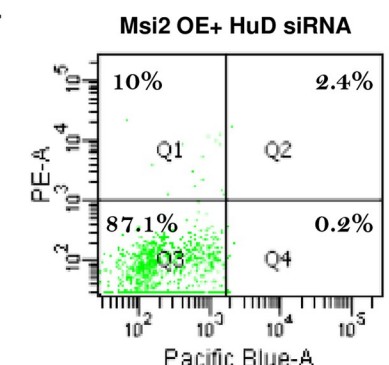

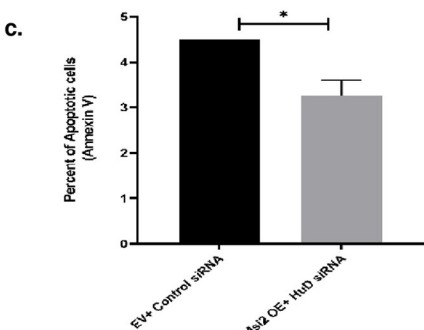

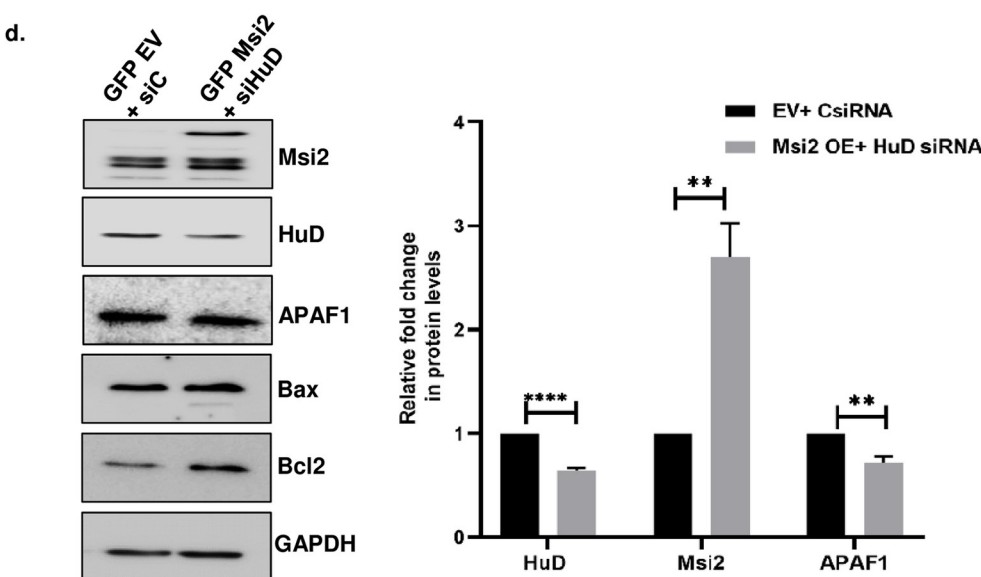

**Fig 5. Simultaneous HuD knockdown and overexpression of Msi2 reduces apoptosis in N2a cells.** pEGFP.C1 EV and pEGFP.C1-Msi2 constructs were transfected in N2a cells, six hours later Control siRNA and HuD siRNA were transfected in these cells. After 48 hours of transfection, the cells were trypsinised and stained with Annexin V conjugated Pacific blue and Propidium iodide. Cells were analyzed using a flow cytometer (Canto) **(A)** control siRNA and EV cells **(B)** HuD siRNA and Msi2 overexpressed cells showed a reduction in apoptosis. **(C)** Bar graph

representing percentage of annexin positive (apoptotic) cells in indicated samples. **(D)** Western blot probed with Msi2, HuD, APAF1, Bcl2, Bax, and GAPDH showing no significant change in ratio of Bax/Bcl2 upon HuD knockdown and Msi2 overexpression. The right panel shows the bar graph representing the relative change in HuD, Msi2, APAF1 and Bax/Bcl2 protein normalized to GAPDH levels between control and HuD knockdown and Msi2 overexpression cells from three independent experiments.

of this molecule is thus highly important. Apoptosis is a key contributor to neurodegenerative diseases such as Alzheimer's disease [33, 46]. Numerous indicators suggest that mitochondrial damage is an early event in the development of Alzheimer's disease [43, 44].

Our findings reveal a crucial alternative pathway for regulating apoptosis that involves the RNA-binding proteins HuD and Msi2. By using RNA immunoprecipitation, we demonstrated the interaction of HuD with Msi2 mRNA. We also found that HuD upregulates the expression of Msi2, which in turn binds to the mRNA of *APAF1*. This regulation impacts the expression of APAF1 the major component of the mitochondrial apoptotic pathway. Knockdown of HuDor Msi2 leads to increased APAF1 protein and Bax/Bcl2 ratio and apoptosis in N2a cells. HuD and Msi2 both affect the mitochondrial apoptotic pathway and act as anti-apoptotic factors. In neuroblastoma, HuD is known to regulate the process of apoptosis by positively regulating the mRNA of the ER shaping molecule ARL6IP1 [47]. ARL6IP1 is known to alter the Bax/Bcl2 ratio independent of APAF1 [48]. Simultaneous knockdown of HuD and overexpression of Msi2 leads to reduced APAF1 levels this could be because of the overexpression Msi2 overcomes the effects of HuD knockdown by directly interacting with *APAF1* RNA and

a.

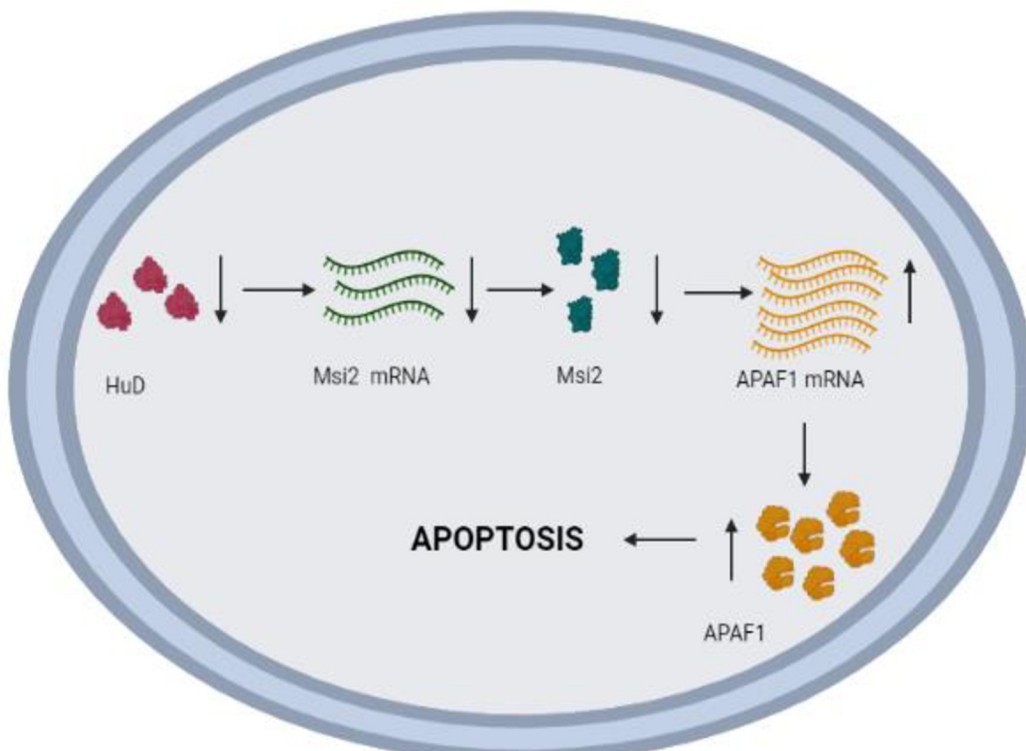

**Fig 6. Schematic overview of apoptotic pathway regulated by HuD and Msi2.** Schematic overview of an alternate apoptotic pathway regulated by HuD via the RNA binding protein Msi2. Msi2 further regulates its target mRNA APAF1 which further influences the process of apoptosis in the N2a cells.

regulating its expression. Thus, we do not observe any significant change in the Bax/Bcl2 ratio while the apoptosis is reduced due to decreased expression of APAF1.

In Alzheimer's disease, the levels of many RNA-binding proteins are altered including the levels of HuD [19]. There is a decrease in the levels of HuD in the hippocampus neurons of the AD brain [8]. HuD knockout in human iPSC-derived neurons can cause an increase in the levels of the Aβ1–42 peptides and the accumulation of phosphorylated Tau, the molecules responsible for the pathogenesis of Alzheimer's disease [49]. All these studies suggest that the RNA-binding protein HuD can be targeted therapeutically. Foxo1 is the transcription factor that suppresses the expression of HuD [3], it is also found that an increase in the expression Foxo1 is correlated to increased incidences of AD [50]. Designing the drug against the Foxo1 ligand that binds to HuD will inhibit the interaction between HuD and Foxo1 and thus relieve the suppression of HuD. However, in-depth studies should be carried out to understand how this could affect the overall signaling pathways regulated by Foxo1. This study unveils an important pathway of apoptosis regulated by the RBPs HuD and Msi2.

## Conclusions

This study expands our understanding of the intricate network of RNA-binding proteins that regulate apoptosis and the fact that HuD acts as a master regulator, controlling the expression of another RNA binding protein Msi2 (Fig 6). Overall, our study suggests that both HuD and Msi2 have an anti-apoptotic role where HuD affects the levels of Msi2, which in turn affects APAF1 levels further influencing the survivability of N2A cells.

## Supporting information

**S1 Fig. Knockdown of HuD in N2a cells causes decrease in the RNA levels of Msi2.** Control siRNA and HuDsi RNAs were transfected in N2a cells and the RNA levels of Msi2 RNA were examined by Real-Time PCR **A)** Graph shows the decrease in the Msi2 RNA levels.
(PPTX)

**S2 Fig. Msi2 overexpression causes decrease in the APAF1 protein levels.** peGFP.C1 EV and peGFP.C1 Msi2 constructs were overexpressed in N2a cells and the protein levels of APAF1 was analyzed by western blotting. (A) Western blot showing protein levels of APAF1 upon Msi2 overexpression. (B) Bar graph represents the relative quantitation of the Msi2 and APAF protein normalized to tubulin levels from three independent western blots.
(PPTX)

**S3 Fig. HuD and Msi2 double knockdown does not causes additive increase in the APAF1 levels and Bax/Bcl2 ratio.** Double knockdown of HuD and Msi2 was carried out in N2a cells using siRNA, the protein levels of APAF1 (A) and ratio of Bax/Bcl2 (B) were analyzed by western blotting. Western blot shows increase in the protein levels of APAF1 upon HuD and Msi2 double knockdown and its graphical representation (right panel).
(PPTX)

**S4 Fig. Sequential overexpression of HuD and Msi2 causes decrease in the Bax/Bcl2 ratio.** HuD and Msi2 were sequentially overexpressed in N2a cells and the Bax/Bcl2 ration was examined by western blotting, **A)** Western blot showing decrease in the Bax/Bcl2 ration upon sequential overexpression of Hud and Msi2. **B)** Graphical representation of the same.
(PPTX)

**S1 Raw image. Raw image of the western blots shown in multiple figures.** The specific figure number is as indicated.
(PDF)

## Acknowledgments

Fellowship to NG, SH by CSIR, India and to RS and GA by DBT, India is greatly acknowledged. We thank Dr. Jomon Joseph for his valuable suggestions and technical help. Special thanks to Dr. Debasish Paul and Dr. Nikhil Ghate for reviewing the manuscript and for providing their valuable inputs.

## Author Contributions

**Conceptualization:** Naina Gaikwad, Vasudevan Seshadri.

**Data curation:** Naina Gaikwad, Rucha Sarwade, Gaurav Agarwal.

**Formal analysis:** Naina Gaikwad, Rucha Sarwade.

**Investigation:** Naina Gaikwad, Gaurav Agarwal.

**Methodology:** Naina Gaikwad, Rucha Sarwade, Sourav Halder.

**Project administration:** Vasudevan Seshadri.

**Supervision:** Vasudevan Seshadri.

**Validation:** Rucha Sarwade, Sourav Halder, Gaurav Agarwal.

**Writing – original draft:** Naina Gaikwad.

**Writing – review & editing:** Vasudevan Seshadri.

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
