## [Decision Letter · Decision Letter 0]

5 Jun 2024

PONE-D-24-08028HuD regulates apoptosis in Neuronal cells by regulating Msi2 expressionPLOS ONE

Dear Dr. Seshadri,

Thank you for submitting your manuscript to PLOS ONE. After careful consideration, we feel that it has merit but does not fully meet PLOS ONE’s publication criteria as it currently stands. Therefore, we invite you to submit a revised version of the manuscript that addresses the points raised during the review process.

We look forward to receiving your revised manuscript.

Kind regards,

Divijendra Natha Reddy Sirigiri

Academic Editor

PLOS ONE

Reviewers' comments:

Reviewer's Responses to Questions

**Comments to the Author**

1. Is the manuscript technically sound, and do the data support the conclusions?

Reviewer #1: Yes

Reviewer #2: No

2. Has the statistical analysis been performed appropriately and rigorously? 

Reviewer #1: No

Reviewer #2: Yes

3. Have the authors made all data underlying the findings in their manuscript fully available?

Reviewer #1: No

Reviewer #2: No

4. Is the manuscript presented in an intelligible fashion and written in standard English?

Reviewer #1: Yes

Reviewer #2: No

5. Review Comments to the Author

Reviewer #1: The researchers show an interesting new role for neuronal RNA-binding protein HuD in apoptosis through interaction with Msi2. Multiple approaches were used to ensure the link between HuD and Msi2 is robust and affects apoptosis.

However, the manuscript is lacking clarity in some areas, and more work could be done to further solidify the authors’ conclusions. Most importantly, the statistical testing, particularly the selected p-value thresholds for statistical significance, need to be revisited.

Copyediting/Writing Clarification:

- Light copyediting is needed – e.g. “interacting and repressing APAF1 transcript” in the abstract should be “interacting with and repressing APAF1 transcript”, and there are other similar grammar errors that should be fixed. The sentence in the results that starts with “Additionally, we examined the levels of APAF1 protein…” contains grammatical errors and appears unfinished.

- APAF1 is described as “the” target of Msi2, and Msi2 is described as “the” target of HuD; these should be corrected to “a” to clarify that each of these RNA-binding proteins has more targets than just those examined in the present study.

- The text should follow conventions for gene and protein names – italicization of genes and mRNAs, no italicization for proteins, and capitalization that reflects whether the gene/protein is from human or mouse. That will make the results much clearer to read and interpret. When referencing prior studies in the introduction and discussion, it should also be made clear what species was used if naming conventions cause it to be ambiguous (e.g., HuD protein is written the same way for both human and mouse).

Methods/Results:

- N2a cells are not defined in the abstract or anywhere else, so those unfamiliar with this cell line will be left in the dark as to the tissue and species.

- More information is needed in regards to the web tool POSTAR. What version was used? Searching on the POSTAR website, there are multiple databases available to use, including one on RNA-binding protein binding sites and another on RNA crosstalk. Which of these was used? How did the authors use this information to determine APAF1 as a target transcript of Msi2?

- Without specific justification for use of one-tailed t-tests, two-tailed tests should be used. A significance threshold of p ≤ 0.1 is also not appropriate for the types of analyses performed here. It’s also quite a large jump from ** = p ≤0.01 to * = p ≤ 0.1 and that makes it difficult to gauge whether the analyses that reach the * = p ≤ 0.1 threshold would be significant with a more appropriate p < 0.05 threshold, particularly if the authors switch to two-tailed t-tests. For example, Fig. 3c would almost certainly not reach significance with more stringent and appropriate statistical testing.

- For western blots, antibody dilutions (e.g. 1:1000) should be listed.

- Bax/Bcl2 ratio and flow cytometry are good complementary measurements of apoptosis. For the flow cytometry figures, the quadrant that shows apoptotic cells should be clarified.

- It is interesting and important to show via HuD knockdown and simultaneous Msi2 overexpression that these two proteins together block apoptosis. The authors perform an experiment with simultaneous HuD and Msi2 knockdown. However, it would be equally relevant to show whether Msi2 overexpression is sufficient to block apoptosis after (not simultaneous) HuD overexpression. This would further reinforce the proposed pathway in Figure 6. On that note, it would also be helpful to see data that HuD knockdown increases Msi2 mRNA in addition to Msi2 protein, and/or that HuD overexpression decreases Msi2 protein in addition to Msi2 mRNA.

Discussion:

- The discussion in general is concise, but thin. This seems to be partly because there is not much prior research on HuD/Msi2 and apoptosis, which is not the fault of the authors. However, more could be said in regards to HuD and apoptosis in Alzheimer’s disease, since this makes up a large portion of the introduction and appears to be a major motivator for the study’s design. Other relevant studies, for example the one on paraneoplastic gut dysmotility, should be better contextualized within the findings of the paper.

Reviewer #2: Ref: PONE-D-24-08028

Title: HuD regulates apoptosis in Neuronal cells by regulating Msi2 expression

Recommendation: MAJOR REVISION

The article investigates the critical role of the RNA-binding protein HuD in neurite outgrowth, neuronal plasticity, and survival, with a particular focus on its influence on the apoptotic pathway in human neuroblastoma cells. The study addresses the enigmatic mechanism by which HuD regulates apoptosis, a fundamental process implicated in neurodegenerative diseases such as Alzheimer's. A significant finding of this research is the identification of Musashi RNA-binding protein 2 (Msi2) as a target transcript of HuD. The authors demonstrate that HuD positively regulates Msi2 expression in N2a neuroblastoma cells. Msi2, in turn, is shown to have diverse mRNA targets and plays a crucial role in regulating the mitochondrial apoptotic pathway by interacting with and repressing the apoptotic protease activating factor 1 (APAF1) transcript.

The study reveals a critical inverse relationship between HuD and apoptosis: a decrease in HuD levels leads to reduced Msi2 expression, increased APAF1 levels, and consequently, apoptosis in N2a cells. This discovery underscores the anti-apoptotic function of both HuD and Msi2 in neuronal cells, highlighting their potential as therapeutic targets for neurodegenerative diseases Overall, this research provides valuable insights into the molecular mechanisms underlying neuronal survival and apoptosis. By elucidating the role of HuD and Msi2 in regulating the apoptotic pathway, the study opens new avenues for therapeutic strategies aimed at mitigating neuronal apoptosis in neurodegenerative diseases.

Major comments:

1. The methodology for N2a cell differentiation as described is insufficiently detailed and lacks critical information necessary for ensuring that the cells were properly differentiated into neuron-like cells. This omission undermines the credibility of the study's findings on the role of HuD and Msi2 in neuronal apoptosis.

2. There is lack of detailed information on the cell line authentication and mycoplasma testing is a significant oversight. Moreover, the description lacks specific details regarding the passage number of the cells used in experiments. Passage number can significantly influence cell behavior, including proliferation rates and gene expression profiles, potentially impacting the study's outcomes. Without this information, it is challenging to assess the reproducibility of the experiments.

3. The methodology described for cloning Msi2 from N2a cells into the pEGFP-C1 vector exhibits several critical issues - the description lacks crucial information about the quality and quantity of RNA extracted from N2a cells.

4. Real time PCR - The use of 18s rRNA as a housekeeping gene for normalization is common; however, the methodology does not justify its selection or validate its stability under the experimental conditions. There is no information on the validation of the primers used for Real-Time PCR. Proper primer validation, including efficiency tests, melting curve analysis, and verification of specific amplification, is essential to ensure accurate quantification of the target mRNAs. Moreover, while the use of 2x SYBR green mix (Biorad) is mentioned, there are no details on the specific real-time PCR instrument used, the reaction conditions (e.g., annealing temperatures, cycle numbers), or the protocol for setting up the reactions. These details are critical for reproducibility and for other researchers to replicate the study.

5. The methodology of western blot specifies the antibodies used but lacks details on the concentrations or dilutions of the primary or secondary antibodies. Moreover, the whole images of wb should also be included into supplementary information.

6. The omission of the study's aim from the end of the Introduction section, replaced by the study's conclusion. Please add the aim of the study instead.

7. At the end of the paper, please add the conclusions.

Minor comments:

1. The introduction section is disorganized. Please rewrite it for clarity and coherence.

2. The discussion section of this paper is disappointing for several reasons. Firstly, it lacks depth and critical analysis – the Authors fail to engage with the broader literature or consider alternative interpretations of their results.

3. Moreover, the discussion fails to address several important questions that arise from the research presented. For instance, the authors mention the implications of their findings for developing therapeutics but do not elaborate on potential strategies or challenges in targeting HuD and Msi2 for clinical applications.

4. Furthermore, the discussion section lacks coherence and organization.

5. The nomenclature of genes and proteins is a standardized system used. Please rewrite it accordingly.

6. The control bars in each figure are missing error bars.

6. PLOS authors have the option to publish the peer review history of their article (what does this mean?). If published, this will include your full peer review and any attached files.

Reviewer #1: No

Reviewer #2: No

---

## [Author Response · Author response to Decision Letter 0]

20 Aug 2024

We thank the reviewers for their thoughtful feedback, comments and valuable suggestions. Below, we have addressed all the comments provided by the reviewers point to point. Reviewer’s comments are shown in italics and our response is shown in blue.

Reviewer #1: Most importantly, the statistical testing, particularly the selected p-value thresholds for statistical significance, need to be revisited.

We have carried out a reanalysis of all the data and corrected it, now we have applied a two-tailed T-Test with the threshold of *p≤0.05 for significance

Copyediting/Writing Clarification: Light copyediting is needed – e.g. “interacting and repressing APAF1 transcript” in the abstract should be “interacting with and repressing APAF1 transcript”, and there are other similar grammar errors that should be fixed. The sentence in the results that starts with “Additionally, we examined the levels of APAF1 protein…” contains grammatical errors and appears unfinished. APAF1 is described as “the” target of Msi2, and Msi2 is described as “the” target of HuD; these should be corrected to “a” to clarify that each of these RNA-binding proteins has more targets than just those examined in the present study. The text should follow conventions for gene and protein names – italicization of genes and mRNAs, no italicization for proteins, and capitalization that reflects whether the gene/protein is from human or mouse. That will make the results much clearer to read and interpret. When referencing prior studies in the introduction and discussion, it should also be made clear what species was used if naming conventions cause it to be ambiguous (e.g., HuD protein is written the same way for both human and mouse). 

We thank the reviewer for pointing out the errors, we have made all the necessary changes. We have also italicized the names of mRNAs with no italicization of protein names as per the convention. We have made it clear also in the introduction which species the study was carried out in.

Methods/Results: N2a cells are not defined in the abstract or anywhere else, so those unfamiliar with this cell line will be left in the dark as to the tissue and species. More information is needed in regards to the web tool POSTAR. What version was used? Searching on the POSTAR website, there are multiple databases available to use, including one on RNA-binding protein binding sites and another on RNA crosstalk. Which of these was used? How did the authors use this information

to determine APAF1 as a target transcript of Msi2? Without specific justification for use of one-tailed t-tests, two-tailed tests should be used. A significance threshold of p ≤ 0.1 is also not appropriate for the types of analyses performed here. It’s also quite a large jump from ** ≤0.01 to * p ≤ 0.1 and that makes it difficult to gauge whether the analyses that reach the * p ≤ 0.1 threshold would be significant with a more appropriate p < 0.05 threshold, particularly if the authors switch to two-tailed t-tests. For example, Fig. 3c would almost certainly not reach significance with more stringent and appropriate statistical testing. For western blots, antibody dilutions (e.g. 1:1000) should be listed. Bax/Bcl2 ratio and flow cytometry are good complementary measurements of apoptosis. For the flow cytometry figures, the quadrant that shows apoptotic cells should be clarified. It is interesting and important to show via HuD knockdown and simultaneous Msi2 overexpression that these two proteins together block apoptosis. The authors perform an experiment with simultaneous HuD and Msi2 knockdown. However, it would be equally relevant to show whether Msi2 overexpression is sufficient to block apoptosis after (not simultaneous) HuD overexpression. This would further reinforce the proposed pathway in Figure 6. On that note, it would also be helpful to see data that HuD knockdown increases Msi2 mRNA in addition to Msi2 protein, and/or that HuD overexpression decreases Msi2 protein in addition to Msi2 mRNA.

We used mouse Neuroblastoma cell line for our study obtained from the brain tissue of the strain A albino mice. They consist of the neuronal and ameboid stem cell morphology. We have updated about the source in the literature. We used POSTAR 2, CLIP database for our study to identify the mRNA that could interact with Msi2. We found that Msi2 RBP had 17 binding sites on the mRNA of APAF1, apart from this the CLIP database also provides a score of the RBP binding to RNA, which was high in this case. We thank the reviewer for raising this point, we have also updated this information in the materials and methods. We also thank the reviewer for pointing out the flaw in the statistical analysis, mentioning * p ≤ 0.1 in the statistical analysis was an error from our side, even before, with the one-tailed T-Test, we had applied the threshold of 0.05. We have carried out a reanalysis of all the data and corrected it, now we have applied a two-tailed T-Test with the threshold of *p≤0.05. 

We have mentioned the dilutions of all the antibodies used for experiments in the materials and methods. We have also named the quadrants that denote the apoptotic cells in the FACS data.

According to the reviewer’s suggestion, we have carried out the sequential overexpression of HuD followed by Msi2 and carried out the western blotting to check the Bax/Bcl2 and we found a significant decrease in the same. We have included the blot in supplementary data (Supplimentary Fig 4). 

In addition we carried out HuD knockdown in N2a cells and assessed the levels of Msi2 RNA using Real-Time PCR, we found a significant decrease in Msi2 RNA levels upon HuD knockdown which supports our previous data that HuD positively regulates the RNA levels of Msi2. We have added this data in the supplementary figures (Supplimentary Fig 1).

Discussion: The discussion in general is concise, but thin. This seems to be partly because there is not much prior research on HuD/Msi2 and apoptosis, which is not the fault of the authors. However, more could be said in regards to HuD and apoptosis in Alzheimer’s disease, since this makes up a large portion of the introduction and appears to be a major motivator for the study’s design. Other relevant studies, for example the one on paraneoplastic gut dysmotility, should be better contextualized within the findings of the paper. 

As per the reviewer’s suggestion, we have elaborated the discussion. We have also discussed about how HuD contributes in the pathogenesis of Alzheimer’s disease. 

Reviewer #2: Ref: PONE-D-24-08028

Major comments: 

1. The methodology for N2a cell differentiation as described is insufficiently detailed and lacks critical information necessary for ensuring that the cells were properly differentiated into neuron-like cells. This omission undermines the credibility of the study findings on the role of HuD and Msi2 in neuronal apoptosis.

We used undifferentiated N2a cells for all our experiments. The undifferentiated N2a cells express low levels of neuronal markers like NeuN and MAP2. Apart from this the undifferentiated N2a cells are known to express the neurofilaments as these are highly committed. But the reviewer has raised an important point for the same reason we have changed the title of the paper to HuD regulates the apoptosis in N2a cells by regulating Msi2. We have also avoided using the term neuronal apoptosis further in the paper.

2. There is lack of detailed information on the cell line authentication and mycoplasma testing is a significant oversight. Moreover, the description lacks specific details regarding the passage number of the cells used in experiments. Passage number can significantly influence cell behavior, including proliferation rates and gene expression profiles, potentially impacting the study’s outcomes. Without this information, it is challenging to assess the reproducibility of the experiments.

We thank the reviewer for pointing this out. We got the Neuro-2A cells from the National Centre for Cell Science, cell repository, all the cell lines are authenticated at NCCS before distribution. The cell lines are also tested for mycoplasma contamination in the NCCS repository before distribution. We used early passage N2a cells for all our experiments.

3. The methodology described for cloning Msi2 from N2a cells into the pEGFP-C1 vector exhibits several critical issues the description lacks crucial information about the quality and quantity of RNA extracted from N2a cells.

We thank the reviewer for pointing out this. We extracted RNA from a single 10cm dish using homemade Trizol. The RNA quality was checked on MOPS gel and the quality was good, as we could see 28s rRNA and 18srRNA as well as their ratio was 2:1. We also verified the quality and quantity using nanodrop, the peak was good and 260/280 ratio was 1.8. We used 1µg of RNA for cDNA synthesis followed by PCR amplification of Msi2 CDS region. We have also updated this information in the materials and methods.

4. Real time PCR - The use of 18s rRNA as a housekeeping gene for normalization is common; however, the methodology does not justify its selection or validate its stability under the experimental conditions. There is no information on the validation of the primers used for Real-Time PCR. Proper primer validation, including efficiency tests, melting curve analysis, and verification of specific amplification, is essential to ensure accurate quantification of the target mRNAs. Moreover, while the use of 2x SYBR green mix (Biorad) is mentioned, there are no details on the specific real-time PCR instrument used, the reaction conditions (e.g., annealing temperatures, cycle numbers), or the protocol for setting up the reactions. These details are critical for reproducibility and for other researchers to replicate the study.

We used 18s rRNA gene as housekeeping for the Real-time experiment as it’s expression was stable under all the experimental conditions. Before using the primers for the Real-Time, we used the same set of primers for RIP experiment, we had also resolved the amplicons on agarose gel. All the primers used for Real-Time had given us specific bands i.e specific amplification on agarose gel. We used epperndorf realpex mastercylcer machine for Real-Time PCR. We have updated the information regarding the annealing temperature, cycle number in the materials and methods. 

5. The methodology of western blot specifies the

antibodies used but lacks details on the concentrations or dilutions of the primary or secondary antibodies. Moreover, the whole images of wb should also be included into supplementary information.

We thank the reviewer for this suggestion. We have updated the information about the dilutions of all the antibodies used for western blotting. We have also provided a file with whole western blot images for the reviewer’s reference.

6. The omission of the study’s aim from the end of the Introduction section, replaced by the study’s conclusion. Please add the aim of the study instead.

We thank the reviewer for pointing this out. We have added the aim of the study in the introduction section.

7. At the end of the paper, please add the conclusions.

We have included the conclusion in the end of the paper.

Minor comments:

1. The introduction section is disorganized. Please rewrite it for clarity and coherence. 

4. Furthermore, the discussion section lacks coherence and organization

 As per the reviewer’s suggestion we have rewritten the introduction for clarity.

2. The discussion section of this, paper is disappointing for several reasons. Firstly, it lacks depth and critical analysis – the Authors fail to engage with the broader literature or consider alternative interpretations of their results. 

As per the reviewer’s suggestion, we have also rewritten the discussion. We have tried our best to make the discussion engaging, we have also discussed the results elaborately.

3. Moreover, the discussion fails to address several important questions that arise from the research presented. For instance, the authors mention the implications of their findings for developing therapeutics but do not elaborate on potential strategies or challenges in targeting HuD and Msi2 for clinical applications.

As per the reviewer’s suggestion, we have written about the potential strategies that can be used for targeting HuD and the challenges.

5. The nomenclature of genes and proteins is a standardized system used. Please rewrite it accordingly.

We thank the reviewer for pointing this out we have changed the RNA names to italics.

6. The control bars in each figure are missing error bars.

All the control were normalized and set as one hence, there is no error bar on the control. The only graphs where the control has error bars are in the FACS data. In this case the control values were taken as it is for comparing with test.

---

## [Decision Letter · Decision Letter 1]

27 Nov 2024

HuD regulates apoptosis in N2a cells by regulating Msi2 expression

PONE-D-24-08028R1

Dear Dr. Seshadri,

We’re pleased to inform you that your manuscript has been judged scientifically suitable for publication and will be formally accepted for publication once it meets all outstanding technical requirements.

Kind regards,

Divijendra Natha Reddy Sirigiri

Academic Editor

PLOS ONE

Additional Editor Comments (optional):

Reviewers' comments:

Reviewer's Responses to Questions

**Comments to the Author**

1. If the authors have adequately addressed your comments raised in a previous round of review and you feel that this manuscript is now acceptable for publication, you may indicate that here to bypass the “Comments to the Author” section, enter your conflict of interest statement in the “Confidential to Editor” section, and submit your "Accept" recommendation.

Reviewer #2: All comments have been addressed

2. Is the manuscript technically sound, and do the data support the conclusions?

Reviewer #2: Yes

3. Has the statistical analysis been performed appropriately and rigorously? 

Reviewer #2: Yes

4. Have the authors made all data underlying the findings in their manuscript fully available?

Reviewer #2: Yes

5. Is the manuscript presented in an intelligible fashion and written in standard English?

Reviewer #2: Yes

6. Review Comments to the Author

Reviewer #2: The authors have thoroughly addressed all my comments and suggestions, significantly improving the clarity and quality of the paper. The research is well-executed, and the results are presented in a comprehensive manner that aligns with the standards of PLOS ONE. I recommend that the manuscript be accepted for publication.

7. PLOS authors have the option to publish the peer review history of their article (what does this mean?). If published, this will include your full peer review and any attached files.

Reviewer #2: No

---

## [Editor Report · Acceptance letter]

5 Dec 2024

PONE-D-24-08028R1 

PLOS ONE

Dear Dr. Seshadri, 

I'm pleased to inform you that your manuscript has been deemed suitable for publication in PLOS ONE. Congratulations! Your manuscript is now being handed over to our production team.

Kind regards, 

on behalf of

Dr. Divijendra Natha Reddy Sirigiri 

Academic Editor

PLOS ONE